# HLA-A*24 Increases the Risk of HTLV-1-Associated Myelopathy despite Reducing HTLV-1 Proviral Load

**DOI:** 10.3390/ijms25136858

**Published:** 2024-06-22

**Authors:** Masakazu Tanaka, Norihiro Takenouchi, Shiho Arishima, Toshio Matsuzaki, Satoshi Nozuma, Eiji Matsuura, Hiroshi Takashima, Ryuji Kubota

**Affiliations:** 1Division of Neuroimmunology, Joint Research Center for Human Retrovirus Infection, Kagoshima University, 8-35-1 Sakuragaoka, Kagoshima 890-8544, Japan; tanakam@m.kufm.kagoshima-u.ac.jp (M.T.);; 2Department of Microbiology, Kansai Medical University, 2-5-1 Shinmachi, Hirakata, Osaka 573-1010, Japan; 3Department of Neurology, Kansai Medical University, 2-5-1 Shinmachi, Hirakata, Osaka 573-1010, Japan; 4Department of Neurology and Geriatrics, Kagoshima University Graduate School of Medical and Dental Sciences, 8-35-1 Sakuragaoka, Kagoshima 890-8544, Japanpine@m.kufm.kagoshima-u.ac.jp (E.M.);

**Keywords:** HTLV-1, HTLV-1-associated myelopathy/tropical spastic paraparesis, HLA-A*24, cytotoxic T lymphocyte, proviral load

## Abstract

Increased human T-cell leukemia virus type 1 (HTLV-1) proviral load (PVL) is a significant risk factor for HTLV-1-associated myelopathy/tropical spastic paraparesis (HAM/TSP). There is controversy surrounding whether HTLV-1-specific cytotoxic T lymphocytes (CTLs) are beneficial or harmful to HAM/TSP patients. Recently, HTLV-1 Tax 301–309 has been identified as an immunodominant epitope restricted to HLA-A*2402. We investigated whether HLA-A*24 reduces HTLV-1 PVL and the risk of HAM/TSP using blood samples from 152 HAM/TSP patients and 155 asymptomatic HTLV-1 carriers. The allele frequency of HLA-A*24 was higher in HAM/TSP patients than in asymptomatic HTLV-1 carriers (72.4% vs. 58.7%, odds ratio 1.84), and HLA-A*24-positive patients showed a 42% reduction in HTLV-1 PVL compared to negative patients. Furthermore, the PVL negatively correlated with the frequency of Tax 301–309-specific CTLs. These findings are opposite to the effects of HLA-A*02, which reduces HTLV-1 PVL and the risk of HAM/TSP. Therefore, we compared the functions of CTLs specific to Tax 11–19 or Tax 301–309, which are immunodominant epitopes restricted to HLA-A*0201 or HLA-A*2402, respectively. The maximum responses of these CTLs were not different in the production of IFN-γ and MIP-1β or in the expression of CD107a—a marker for the degranulation of cytotoxic molecules. However, Tax 301–309-specific CTLs demonstrated 50-fold higher T-cell avidity than Tax 11–19-specific CTLs, suggesting better antigen recognition at low expression levels of the antigens. These findings suggest that HLA-A*24, which induces sensitive HTLV-1-specific CTLs, increases the risk of HAM/TSP despite reducing HTLV-1 PVL.

## 1. Introduction

Human T-cell leukemia virus type 1 (HTLV-1) is a human retrovirus that preferentially infects CD4^+^ lymphocytes in vivo [1]. Once infected, the virus undergoes reverse transcription and integrates into the host genome, forming a provirus [2]. HTLV-1 infection is estimated to affect 10–20 million people worldwide. While HTLV-1 infection persists throughout life, the majority of infected individuals remain asymptomatic. The lifetime risk of developing adult T-cell leukemia (ATL) in people with HTLV-1 is estimated to be 2–7%, whereas the risk of developing HTLV-1-associated myelopathy/tropical spastic paraparesis (HAM/TSP) is 0.25–3% [3,4,5].

HAM/TSP is an inflammatory disease of the spinal cord characterized by the infiltration of inflammatory cells, primarily CD8^+^ T cells, into the perivascular area [6]. Patients typically exhibit central nervous system (CNS) lesions leading to a progressive weakness of the lower extremities with spasticity, mild sensory disturbance, and urinary incontinence [7]. HAM/TSP patients are found to have a 16-fold higher HTLV-1 proviral load (PVL) in peripheral blood mononuclear cells (PBMCs) compared to asymptomatic carriers (ACs) [8,9], suggesting that increased PVL is a strong risk factor for HAM/TSP. Moreover, the progression of HAM/TSP is also associated with an increase in PVL [10]. One notable feature of the immune response in HAM/TSP patients is the significant increase in the number of HTLV-1-specific CD8^+^ cytotoxic T lymphocytes (CTLs) in PBMCs compared to ACs [11]. These CTLs primarily recognize the HTLV-1 Tax protein, a trans-activator of HTLV-1 gene expression, and various host genes [12]. Upon recognition of viral antigens, these CTLs produce elevated levels of proinflammatory cytokines such as interferon (IFN)-γ and tumor necrosis factor (TNF)-α [13,14]. Analysis of human leukocyte antigen (HLA)-A*02/HTLV-1 Tax 11–19 dimers revealed an even higher frequency of Tax 11–19-specific CD8^+^ CTLs in the cerebrospinal fluid (CSF) compared to PBMCs in HAM/TSP patients [15]. Furthermore, HTLV-1 DNA and mRNA have been detected in infiltrating CD4^+^ T cells in the spinal cords of HAM/TSP patients, indicating the involvement of HTLV-1-specific CTLs in the pathogenesis of HAM/TSP [16,17,18].

Several associations between HLA and the outcome of viral infections have been reported. For example, HLA-B*27 and -B*57 are identified as protective in human immunodeficiency virus type-1 (HIV-1) infection, while B*35 and B*53 act as susceptible [19,20]. In the context of HTLV-1 infection, HLA-A*02 and -Cw08 are identified as protective for HAM/TSP, whereas HLA-B*5401 increases susceptibility to the disease [21,22]. CD8^+^ CTLs play a crucial role in controlling viral replication by recognizing viral antigens and killing virus-infected cells via HLA class I restriction. Therefore, the relationship between HLA and the outcome of infectious diseases may depend on CD8^+^ CTL responses [20,23].

While CD8^+^ CTLs are traditionally viewed as protective against viral diseases by clearing virus-infected cells, recent studies have shown that CTL responses to pathogens can sometimes lead to harmful immunopathology in the host [24,25]. The role of vigorous HTLV-1-specific CTL responses observed in HAM/TSP patients in the development of HAM/TSP remains controversial [11,21]. However, some studies suggest that CTL responses may be beneficial to the host. For instance, HLA-A*02 is associated with a reduced risk of HAM/TSP, and HLA-A*02-restricted HTLV-1 Tax 11–19-specific CTL responses are shown to decrease HTLV-1 PVL [21]. Recently, HTLV-1 Tax 301–309 has been identified as an immunodominant epitope restricted to HLA-A*2402 [26,27]. In this study, we aimed to investigate whether HLA-A*24 also reduces HTLV-1 PVL and the risk of HAM/TSP. Surprisingly, we found that despite a reduction in HTLV-1 PVL, HLA-A*24 increased the risk of HAM/TSP, contrasting with the effects of HLA-A*02. Furthermore, we compared Tax-specific CTL responses between HLA-A*02 and -A*24-positive HAM/TSP patients and found that HLA-A*24-restricted Tax-specific CTLs exhibited higher T-cell avidity than HLA-A*02-restricted Tax-specific CTLs.

## 2. Results

### 2.1. Allele Frequency of HLA-A*24 and HTLV-1 PVL

The frequency of the HLA-A*24 allele in HAM/TSP patients and ACs was 72.4% and 58.7%, respectively (Table 1). The frequency was significantly higher in HAM/TSP patients (*p* = 0.017), with the odds ratio of HLA-A*24 occurrence in HAM/TSP being 1.84. HTLV-1 PVLs were quantified using quantitative PCR. The PVL was 1.7-fold lower in HLA-A*24-positive HAM/TSP patients compared to negative patients, with statistical significance (*p* = 0.009, Table 2). The PVL in HLA-A*24-positive ACs was lower than that in negative ACs, but the difference was not significant (*p* = 0.148).

### 2.2. Detection of HLA-A*24-Restricted HTLV-1 Tax 301–309-Specific CTLs

Using the HLA-A*2402/Tax 301–309 pentamer, we determined the frequency of Tax 301–309-specific CTLs in CD8-high cells in HLA-A*24-positive HAM/TSP patients or ACs. Figure 1A depicts a representative flow cytometry analysis. The average frequency was 4.34% and 1.68% in HAM/TSP patients and ACs, respectively (Figure 1B). Although the frequency was higher in HAM/TSP patients compared to ACs, the difference was not significant (*p* = 0.092).

### 2.3. Frequency of Tax 301–309-Specific CTLs Negatively Correlated with HTLV-1 PVL

HTLV-1 PVLs were plotted against the frequency of Tax 301–309-specific CTLs in HLA-A*24-positive patients (Figure 2). The PVLs exhibited significantly negative correlations with CTL frequencies in HAM/TSP patients (*p* = 0.0196), ACs (*p* = 0.0059) and HTLV-1-infected individuals, encompassing both groups (*p* = 0.0002).

### 2.4. In Silico Analysis of Tax Epitope/HLA Binding Affinity

HTLV-1 Tax antigen/HLA binding affinities were predicted using the internet programs NetMHCpan-4.1 and NetCTL-1.2 (Table 3). The NetMHC scores were 0.982 and 0.925, and the NetCTL scores were 1.474 and 1.890 for Tax 11–19/HLA-A*0201 and Tax 301–309/HLA-A*2402, respectively. Although the NetMHC prediction score of Tax 301–309/HLA-A*2402 was slightly lower than that of Tax 11–19/HLA-A*0201, it was slightly higher than that of NetCTL.

### 2.5. CTL Function in HTLV-1 Tax-Specific CTLs Restricted to HLA-A*02- or HLA-A*24 in HAM/TSP Patients

To compare CTL functions of HLA-A*02-restricted Tax 11–19-specific CTLs and HLA-A*24-restricted Tax 301–309-specific CTLs, patients were selected from either HLA-A*02-positive or HLA-A*24-positive HAM/TSP patients, with relatively high CTL frequencies over 0.5%, and available PBMCs. The frequency of Tax-specific CTLs in CD8^+^ cells in each group was 6.53 +/− 7.85% (mean +/− SD) and 6.61 +/− 4.21%, respectively (*p* = 0.40). Figure 3A shows a representative flow cytometric analysis for the tetramer detection of CTLs without culture and for CTL functions after antigenic stimulation. The maximum CTL responses, including IFN-γ production, MIP-1β production, and degranulation measured by CD107a expression, were not different between HLA-A*02- and HLA-A*24-positive HAM/TSP patients (Figure 3B). The mean fluorescent intensity of IFN-γ, MIP-1β, and CD107a of the positive cells did not differ between the two groups.

### 2.6. Functional T-Cell Avidity of HLA-A*02- or HLA-A*24-Restricted Tax-Specific CTLs in HAM/TSP Patients

To compare the functional T-cell avidity of HLA-A*02-restricted Tax 11–19-specific CTLs and HLA-A*24-restricted Tax 301–309-specific CTLs, patients were selected from either HLA-A*02-positive or HLA-A*24-positive HAM/TSP patients, with relatively high CTL frequencies over 0.5%. The average frequency of Tax-specific CTLs from three patients in each group was 1.79 +/− 1.11% (mean +/− SD) and 3.45 +/− 1.21%, respectively. CTL responses were assessed by IFN-γ production induced by serially diluted Tax peptides. Figure 4A shows a representative flow cytometric analysis of IFN-γ detection at a given concentration of Tax 11–19 peptide using PBMCs from an HLA-A*02-positive patient. Figure 4B shows standardized titration curves in Tax 11–19- or Tax 301–309-specific CTLs. The EC50 in Tax 11–19-specific and Tax 301–309-specific CTLs was 31 nM and 0.6 nM, respectively. The functional T-cell avidity of Tax 301–309-specific CTLs was 50-fold higher than that of Tax 11–19-specific CTLs.

## 3. Discussion

We demonstrated that the HTLV-1 PVL was lower in HLA-A*24-positive HAM/TSP patients than in A*24-negative patients (Table 2), and the frequency of HLA-A*24-restricted HTLV-1 Tax 301–309-specific CTLs negatively correlated with the HTLV-1 PVL in both HAM/TSP patients and ACs (Figure 2). These results suggest that Tax 301–309-specific CTLs contribute to the elimination of HTLV-1-infected cells in vivo. However, the allele frequency of HLA-A*24 was higher in HAM/TSP patients than in ACs, suggesting that it increases the risk of HAM/TSP (Table 1).

In viral infections such as HIV-1, hepatitis B virus, and hepatitis C virus infection, viral load is an important determinant of the outcome of infection. In HTLV-1 infection, high HTLV-1 PVL is a strong risk factor for HAM/TSP [9,28]. In this study, HTLV-1 PVL was decreased in HLA-A*24-positive HAM/TSP patients compared to negative patients, while the allele frequency of HLA-A*24, which presents Tax 301–309, an immunodominant epitope for CTLs, was increased in HAM/TSP patients compared to ACs. Therefore, HLA-A*24 has a dual effect of increasing the risk of HAM/TSP and decreasing HTLV-1 PVL, suggesting that HLA-A*24 is a more critical factor than increased HTLV-1 PVL in the development of HAM/TSP in HLA-A*24-positive individuals. Generally, in infectious diseases, disease susceptibility to a particular HLA class I allele is strongly associated with CTL responses restricted to that HLA allele [20,23], and CTLs play a pivotal role in controlling viral infections by killing infected cells. Our results suggest that HTLV-1-specific CTLs reduce the PVL, though they may be harmful to the host, at least in HLA-A*24-positive individuals with HAM/TSP, because HLA-A*24 increases the risk of HAM/TSP.

It has been shown that in viral infections, CD8^+^ CTLs protect the host by killing virus-infected cells. However, recent studies demonstrate that CTL responses to a pathogen sometimes cause immunopathological harm to the host [24,25]. For example, in a mouse model of CNS viral infection with lymphocytic choriomeningitis virus, the depletion of CD8^+^ T cells prevented the animals from dying [29]. In another study, highly activated CD8^+^ T cells in the brain were associated with early CNS dysfunction in simian immunodeficiency virus infection [30]. In mice infected with mouse hepatitis virus, CD8^+^ T cells can cause CNS demyelination in the absence of viral antigen in the CNS, but only if these cells are specifically activated [31]. A recent study on animals infected with Zika virus and displaying paralysis revealed that disease severity does not correlate with brain Zika virus titers but rather with the infiltration of bystander activated CD8^+^ T cells [32]. Moreover, the depletion of CD8^+^ cells prevented the paralysis, suggesting that bystander-activated CD8^+^ cells play a more pivotal role in neuropathological processes in the CNS than viral load. The CNS pathology of HAM/TSP shows a marked accumulation of inflammatory cells around small vessels with a predominance of CD8^+^ lymphocytes. The frequency of HTLV-1-specific CTLs in the circulation and CSF is higher in HAM/TSP patients than in ACs, and the CTLs accumulate in the CNS [11,14,18,33]. Moreover, in a short-term culture of PBMCs from HAM/TSP patients, HTLV-1-specific CTLs produce proinflammatory cytokines, including IFN-γ and TNF-α, by recognizing HTLV-1 antigens expressed on autologous HTLV-1-infected cells during culture [14]. Additionally, HTLV-1 DNA, mRNA, and proteins are found in infiltrating CD4^+^ cells in the brain, but not in resident neural cells, in HAM/TSP patients [16,17,18]. These data support the hypothesis that an interaction between HTLV-1-specific CTLs and virus-infected CD4^+^ cells infiltrating the CNS from the periphery could cause inflammation and immunopathology in the CNS. This inflammation may result in bystander damage to CNS resident cells in HAM/TSP [18,34].

HLA-A*24 increases the risk of HAM/TSP but decreases HTLV-1 PVL, which contrasts with the effects of HLA-A*02. It has been reported that the allele frequency of HLA-A*02 is lower in HAM/TSP patients than in ACs, and the PVL in HLA-A*02-positive patients is lower than that in A*02-negative patients [21]. These results suggest that HLA-A*02 reduces the risk of HAM/TSP, probably due to the reduction of HTLV-1 PVL by HLA-A*02-restricted Tax 11–19-specific CTLs. Therefore, we hypothesized that HLA-A*24-restricted Tax 301–309-specific CTLs have a different CTL function from HLA-A*02-restricted Tax 11–19-specific CTLs and more strongly contribute to the immunopathology in the CNS of HAM/TSP patients. We first performed in silico analysis of Tax epitope/HLA binding affinity for both CTLs (Table 3). However, the difference was not evident because the prediction score of Tax 301–309/HLA-A*2402 was slightly lower than that of Tax 11–19/HLA-A*02 according to the NetMHC program but slightly higher according to the NetCTL program. Therefore, we compared the CTL function of Tax 11–19- and Tax 301–309-specific CTLs by IFN-γ or MIP-1β production and CD107a expression (Figure 3). The maximum responses to the relevant antigens were not different between the two CTL groups from HAM/TSP patients. Next, we compared the functional T-cell avidity of the two CTLs (Figure 4). Although the avidities were similar among patients with the same HLA, Tax 301–309-specific CTLs showed 50-fold higher T-cell avidity than Tax 11–19-specific CTLs. It has been reported that the CTL responses with high T-cell avidity are associated with low viral loads in HTLV-1 and HIV infections [22,35]. Therefore, our results suggest that Tax 301–309-specific CTLs more efficiently recognize HTLV-1-infected cells. Interestingly, the result was consistent with the in silico analysis of the NetCTL program, in which the prediction score was higher in Tax 301–309/HLA-A*2402 than in Tax 11–19/HLA-A*0201. This may be due to the fact that the score is predicted by several factors for antigen presentation process, not only by a simple antigen/HLA binding affinity shown in the NetMHC program [36].

The reason Tax 11–19-specific CTLs are beneficial to the host while Tax 301–309-specific CTLs are harmful, even though both reduce the PVL, remains unclear. One possible mechanism is as follows: although the expression of viral proteins in HTLV-1-infected cells is at a trace level in the peripheral blood of HAM/TSP patients and ACs, the viral proteins are detectable in infiltrating CD4^+^ cells in the CNS from HAM/TSP patients [18]. Therefore, infiltrating HTLV-1-specific CTLs in the CNS may recognize the viral antigens expressed on the infected cells. In this context, if infiltrating cells infected with HTLV-1 express low levels of viral antigens, Tax 301–309-specific CTLs restricted to HLA-A*24 may recognize the antigens more efficiently than Tax 11–19-specific CTLs restricted to HLA-A*02, resulting in stronger inflammation and leading to the immunopathology of HAM/TSP. However, further study is needed to clarify the exact reason through a more detailed investigation, including pathology, comparing viral protein expression levels and the degree of inflammation in the CNS between HLA-A*02 and HLA-A*24 HAM/TSP patients.

## 4. Materials and Methods

### 4.1. Subjects

A total of 152 HAM/TSP patients and 155 asymptomatic HTLV-1 carriers (ACs) were enrolled for this study (Appendix A). All participants were residents of Kagoshima prefecture, located in southern Kyushu, a region known for its high prevalence of HTLV-1 in Japan. HTLV-1 infection was confirmed serologically via the particle agglutination method followed by PCR verification. Diagnosis of HAM/TSP was made in accordance with WHO guidelines. Blood samples were collected from participants after obtaining informed consent, and PBMCs were isolated and stored in liquid nitrogen until use. The study was approved by the Institutional Ethics Committee of Kagoshima University.

### 4.2. HLA Typing

Genomic DNA was extracted from PBMCs of all participants. HLA-A*02 and HLA-A*24 typing was performed using polymerase chain reaction (PCR) with sequence-specific primers, as previously described [37]. Briefly, DNA was extracted from PBMCs, and 50 ng of DNA was subjected to PCR with sequence-specific primer sets for HLA-A*02 (#292 and #170) and for an internal control (#63 and #64), using AmpliTaq Gold 360 master mix (Applied BioSystems, Tokyo, Japan). The cycling program included denaturation at 96 °C for 25 s, annealing at 70 °C for 45 s, and extension at 72 °C for 45 s for the first 5 cycles; denaturation at 96 °C for 25 s, annealing at 65 °C for 50 s, and extension at 72 °C for 45 s for the next 21 cycles; and denaturation at 96 °C for 25 s, annealing at 55 °C for 60 s, and extension at 72 °C for 120 s for the final 4 cycles. The PCR products were electrophoresed in a 2% agarose gel. The product sizes were 557 bp and 796 bp, respectively.

### 4.3. Quantification of HTLV-1 PVL

Quantitative PCR was conducted as per previously described methods [9]. Briefly, 100 ng of DNA extracted from PBMCs underwent quantitative PCR using HTLV-1 tax primers, β-actin primers, and TaqMan probes (Applied Biosystems, Tokyo, Japan). The assay was performed in triplicate, and copy numbers were determined via standard curves. PVL per 10^4^ PBMCs was calculated using the formula: (mean of tax copy numbers)/(mean of β-actin copy numbers) × 2 × 10^4^.

### 4.4. Detection of HLA-A*24-Restricted HTLV-1 Tax 301–309-Specific CTLs in PBMCs

HTLV-1 Tax 301–309 (SFHSLHLLF) is a known immunodominant epitope restricted to HLA-A*2402 [26,38]. PBMCs from 20 HAM/TSP patients and 19 ACs underwent flow cytometric analysis. Then, 1 × 10^6^ cells were stained with phycoerythrin (PE)-conjugated HLA-A*2402/Tax 301–309 pentamer (Proimmune, Oxford, UK) and PE-Cy5 (PC5)-conjugated anti-CD8 antibody (Beckman Coulter, Tokyo, Japan) for 20 min at room temperature. An HLA-A*2402/HIV-1 Gag pentamer was used as a relevant control. Flow cytometry was performed using an Epics-XL flow cytometer (Beckman Coulter, Tokyo, Japan). After gating lymphocytes based on forward and side scatters, CTL frequency in CD8-high cells was determined (Appendix A).

### 4.5. In Silico Analysis of Tax Epitope/HLA Binding Affinity

To compare the binding affinity of Tax 11–19/HLA-A*0201 and Tax 301–309/HLA-A*2402, we utilized the NetMHCpan-4.1 server (https://services.healthtech.dtu.dk/services/NetMHCpan-4.1/, accessed on 16 April 2022) and the NetCTL-1.2 server (https://services.healthtech.dtu.dk/services/NetCTL-1.2/, accessed on 16 April 2022) [39,40]. The binding affinity scores have no units. The NetMHCpan 4.1 score ranks the predicted binding strength of the peptide to MHC as a percentile. The NetCTL 1.2 score integrates the predictions of proteasomal cleavage, TAP transport, and MHC binding into a single score. Higher values indicate a higher likelihood that the peptide will be an effective CTL epitope.

### 4.6. CTL Function in HTLV-1 Tax-Specific CTLs Restricted to HLA-A*02- or HLA-A*24

We chose IFN-γ and MIP-1β as the cytokine and chemokine, respectively, because our preliminary experiments indicated that these proteins were the most sensitive. For a precise evaluation of CTL function, we initially selected HAM/TSP patients with a high frequency of Tax 11–19- or 301–309-specific CTLs exceeding 0.5%. Subsequently, eight patients were randomly chosen from those positive for HLA-A*02 or HLA-A*24, respectively. PBMCs were thawed and washed twice. Then, 2 × 10^6^ cells underwent staining with HLA-A*0201/Tax 11–19 or HLA-A*2402/Tax 301–309 tetramer (MBL, Nagoya, Japan) and anti-CD8 antibody to determine the frequency of Tax-specific CTLs in CD8^+^ cells without culture. CTL frequencies were determined without culture, as T-cell receptor and CD8 molecule expression are down-regulated after antigenic stimulation. Notably, in the absence of an exogenous Tax peptide, some HTLV-1-specific CTLs produce IFN-γ in response to viral proteins expressed on autologous HTLV-1-infected cells during culture. To minimize this production, brefeldin A, which inhibits protein transport from the endoplasmic reticulum to the Golgi apparatus [41], was added at the beginning of the culture. The remaining cells were treated with 1 µM (a concentration inducing a saturated response in either Tax 11–19- or Tax 301–309-specific CTLs in preliminary experiments) of the corresponding Tax 11–19 (LLFGYPVYV) or Tax 301–309 (SFHSLHLLF) peptide for 30 min, followed by washing and distribution into three wells of a culture plate. The 6 × 10^6^ cells were then cultured in the presence of 5 µg/mL of brefeldin A (Sigma-Aldrich, Tokyo, Japan) for 5 h in a CO_2_ incubator. The first and second sets of cells (2 × 10^6^ cells) were used for detecting IFN-γ and macrophage inflammatory protein (MIP)-1β, respectively. The third set of cells (2 × 10^6^ cells) received FITC-conjugated CD107a antibody (BD, Tokyo, Japan) during culture. CD107a is expressed on the cell surface during the degranulation of cytolytic molecules and correlates with cytolytic activity [42,43]. Following harvesting, the first and second sets of cells were stained for intracellular IFN-γ and MIP-1β, respectively, and then for surface CD8. The third set of cells was stained for surface CD8. Frequencies of IFN-γ-, MIP-1β-, and CD107a-positive cells in CD8^+^ cells were determined by flow cytometry (Appendix A). CTL responses of IFN-γ production, MIP-1β production, and CD107a expression were calculated as follows: (frequency of positive cells)/(frequency of tetramer-positive cells without culture) × 100%.

### 4.7. Functional T-Cell Avidity Assay

To evaluate functional T-cell avidity, we performed an antigen titration assay as described previously [44]. Three patients with high frequencies of Tax 11–19- or three patients with high frequencies of 301–309-specific CTLs (>0.5%) were randomly selected from HLA-A*02- or HLA-A*24-positive HAM/TSP patients, respectively. Then, 1 × 10^6^ cells were incubated with serially diluted peptides for 30 min, followed by washing and culturing for 5 h in the presence of brefeldin A. Subsequently, cells were harvested and stained for intracellular IFN-γ and surface CD8 molecules. Finally, IFN-γ-positive cells in CD8^+^ cells were determined by flow cytometry, and the values were normalized by subtracting the frequency without peptides to minimize background. The half-maximal effective concentration (EC50) was calculated.

### 4.8. Statistical Analysis

The chi-square test was used to analyze HLA-A*24 allele frequency. The Mann–Whitney U-test was employed to compare CTL frequency, PVL, IFN-γ production, MIP-1β production, and CD107a expression. Spearman’s rank correlation test was used to analyze the relationship between HTLV-1 PVL and CTL frequency. A *p*-value less than 0.05 was considered statistically significant.

## Figures and Tables

**Figure 1 ijms-25-06858-f001:**
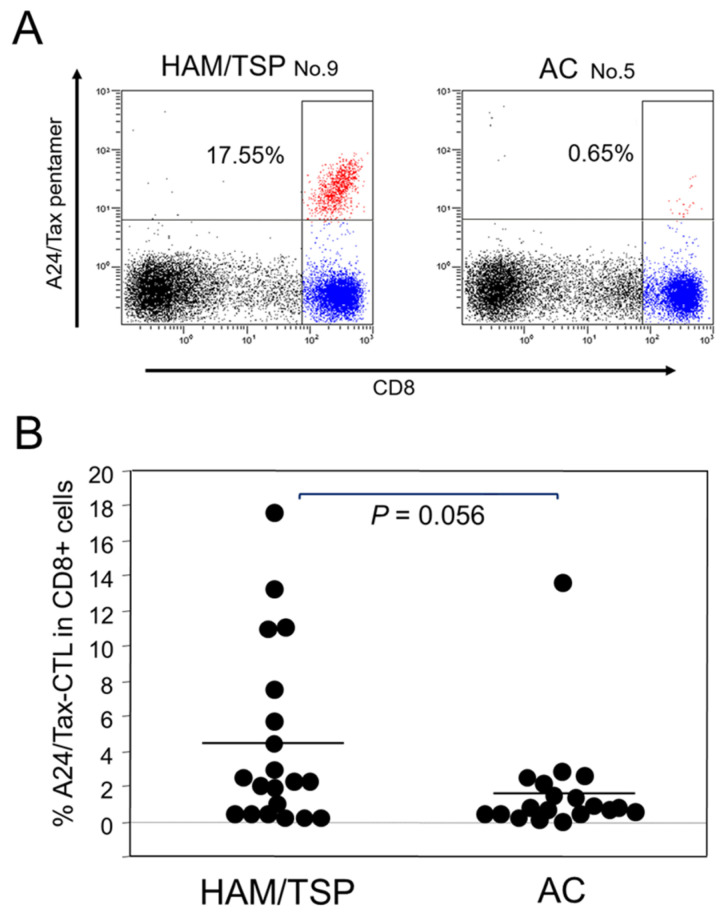
Detection of HLA-A*24-restricted Tax 301–309-specific CTLs in PBMCs of HAM/TSP patients and ACs. (**A**). Representative staining of HLA-A*2402/Tax 301–309 pentamer. The numbers in the figures indicate the percentage of HLA-A*2402/Tax 301–309 pentamer-positive cells in CD8^+^-high cells. The blue and red dots indicate CD8 high cells and pentamer positive cells, respectively. (**B**). Frequency of HLA-A*24-restricted Tax 301–309-specific CTLs in HAM/TSP patients (N = 20) and ACs (N = 19). The average frequencies are 4.34% and 1.68% in HAM/TSP patients and ACs, respectively. There is no significant difference between the patient groups according to the Mann–Whitney U-test, two-tailed.

**Figure 2 ijms-25-06858-f002:**
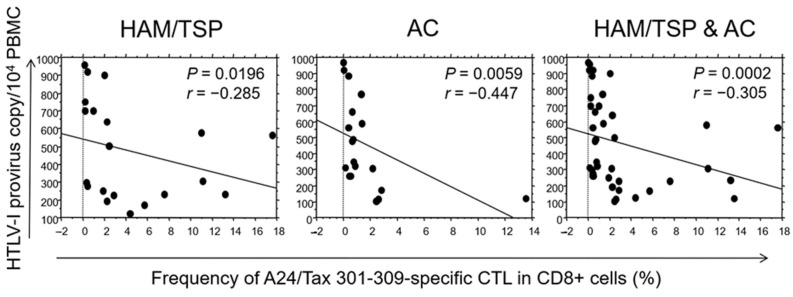
The frequency of HLA-A*24-restricted Tax 301–309-specific CTLs negatively correlates with HTLV-1 PVL. The HTLV-1 PVL were plotted against the frequency of HTLV-1 Tax 301–309-specific CTLs in HAM/TSP patients (N = 20), ACs (N = 19), and HTLV-1-infected individuals, including both groups. The frequency negatively correlates with PVL among all groups, with significances determined by Spearman’s rank correlation test.

**Figure 3 ijms-25-06858-f003:**
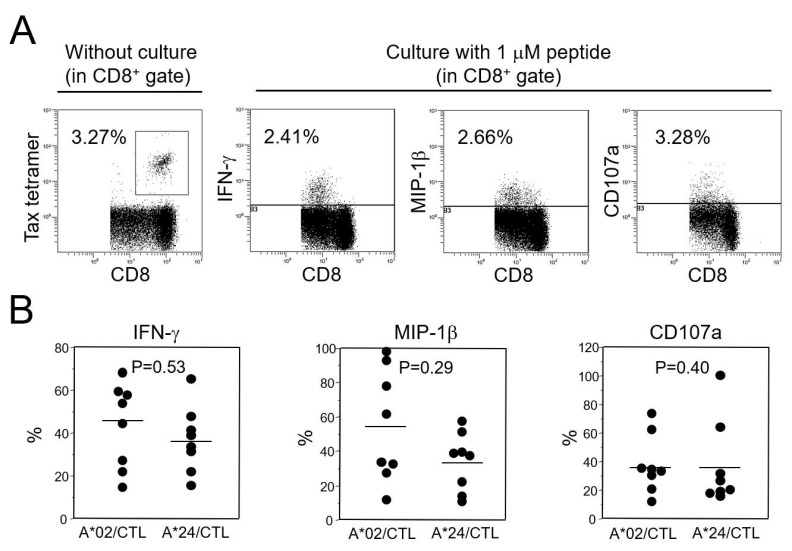
Detection of IFN-γ, MIP-1β or CD107a-positive cells in HLA-A*02 or A*24-restricted CTLs (**A**). Representative detection of HTLV-1 Tax 11–19-specific CTLs in an HLA-A*02-positive patient with HAM/TSP. The frequency of Tax 11–19-specific CTLs in CD8^+^ cells was determined using HLA-A*02/Tax 11–19 tetramer without culture. The remaining cells were cultured in the presence of 1 μM Tax 11–19 peptide for 5 h. IFN-γ, MIP-1β, and CD107a-positive cells were detected. The numbers in the figures indicate the percentage of positive cells in CD8^+^ cells. (**B**). Each symbol indicates the percentage of IFN-γ, MIP-1β or CD107a-positive cells in HLA/Tax antigen tetramer+ cells in either HLA-A*02-positive (N = 8) or HLA-A*24-positive (N = 8) HAM/TSP patients. There are no significant differences between the two groups according to the Mann–Whitney U-test, two-tailed.

**Figure 4 ijms-25-06858-f004:**
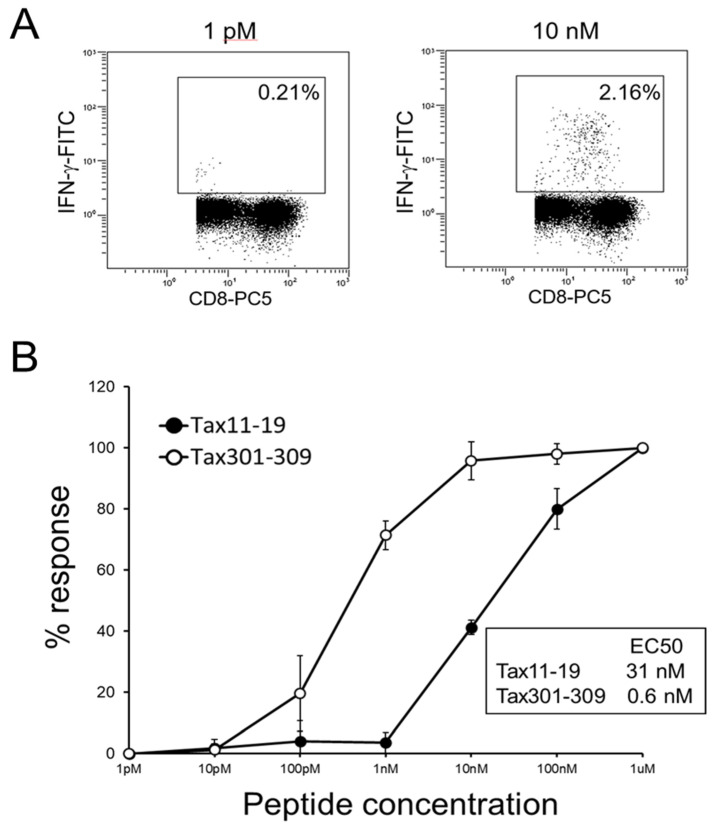
Functional T-cell avidity assay with serially diluted antigen. PBMCs from three HLA-A*02-positive or three HLA-A*24-positive HAM/TSP patients were stimulated with serially diluted peptide, and the frequencies of IFN-γ-positive cells in CD8^+^ cells were determined. (**A**). Representative flow cytometric analysis of IFN-γ-positive cells using PBMCs from an HLA-A*24-positive patient with HAM/TSP. The numbers in the figures indicate the percentage of IFN-γ-positive cells in CD8^+^ cells in the presence of 1 pM or 10 nM peptide. (**B**). After subtraction of the frequency of IFN-γ-positive cells without peptide, the values were standardized to the maximum response, which are plotted against the peptide concentration. The half-maximal effective concentration (EC50) of Tax 11–19 peptide and Tax 301–309 peptide are 31 nM and 0.6 nM, respectively.

**Table 1 ijms-25-06858-t001:** Frequency of HLA-A*24 in HAM/TSP patients and ACs.

	A*24+	A*24−
HAM/TSP	72.4%	27.6%
(*n* = 152)	(110)	(42)
ACs	58.7%	41.3%
(*n* = 155)	(91)	(64)

The numbers in the parentheses indicate the numbers of cases. The *p*-value is 0.017 in the two-tailed chi-square test, and the odds of HLA-A*24 to HAM/TSP is 1.84.

**Table 2 ijms-25-06858-t002:** Effect of HLA-A*24 on HTLV-1 proviral load in HAM/TSP patients and ACs.

Status	HLA	Number of Cases	Proviral Load ^a^	*p* Value ^b^
HAM/TSP	A*24+	110	325 ± 422	0.009
HAM/TSP	A*24−	42	550 ± 506	
ACs	A*24+	91	142 ± 206	0.148
ACs	A*24−	64	198 ± 255	

^a^: HTLV-1 proviral load (copy/10^4^ PBMCs). The value is expressed by mean +/− SD. ^b^: The HTLV-1 proviral loads in HLA-A*24-positive and -negative group are compared by Mann–Whitney U-test in HAM/TSP patients or ACs group.

**Table 3 ijms-25-06858-t003:** Prediction score of Tax epitope/HLA binding affinity.

	Internet Program	
Tax Epitope/HLA	NetMHCpan-4.1	NetCTL-1.2
Tax11–19/HLA-A*0201	0.982	1.474
Tax301–309/HLA-A*2402	0.925	1.890

The binding affinity of epitopes to HLA was predicted using either NetMHCpan-4.1 or NetCTL-1.2 internet programs. Higher scores indicate higher affinities.

## Data Availability

The data presented in this study are available on request to the corresponding author.

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
