# Peer review of "HLA-A*24 Increases the Risk of HTLV-1-Associated Myelopathy despite Reducing HTLV-1 Proviral Load"

_ijms, 2024, doi:10.3390/ijms25136858_

Round 1
Reviewer 1 Report
Comments and Suggestions for Authors
Tanaka et al. report here that HLA-A*24 increases the risk of HAM/TSP despite reducing the HTLV-1 PVL - this is in contrast to HLA-A*02. The work presented here is of high quality and importance. This work illustrates that not only should HTLV-1 PVL be taken into account when following up patients living with HTLV-1, but also the HLA genotype should be determined to more accurately predict the onset of/chances of developing HAM/TSP.
For the in silico prediction of Tax epitope/HLA binding affinity: what are the units and are the differences described significant? Please explain.
I only have two minor comments in the discussion:
* line 268, to avoid confusion - please change to "...than in A*24-negative patients..."
* line 316, again to avoid confusion - please change to "...than that in A*02 negative patients."
Author Response
Reviewer 1
For the in silico prediction of Tax epitope/HLA binding affinity: what are the units and
are the differences described significant? Please explain.
Answer: Thank you for your comment.
The binding affinity scores have no units. The NetMHCpan 4.1 score ranks the
predicted binding strength of the peptide to MHC as a percentile. The NetCTL 1.2 score
integrates the predictions of proteasomal cleavage, TAP transport, and MHC binding
into a single score. Higher values indicate a higher likelihood that the peptide will be an
effective CTL epitope. We could not evaluate the significance between the scores.
These details are described in the text on page 3, line 131.
* line 268, to avoid confusion - please change to "...than in A*24-negative patients..."
Answer: Thank you for your comment.
We have changed as you suggested in line 281.
* line 316, again to avoid confusion - please change to "...than that in A*02 negative
patients."
Answer: Thank you again.
We have replaced as you suggested in line 329.
Reviewer 2 Report
Comments and Suggestions for Authors
This is an exciting study about the relevance of the HLA-A alleles on the HTLV-1-associated HAM/TSP. The article is well-written, and most of the section has enough details, but others need improvement.
Clinical data from patients must be included. Are differences in sex, age, or another variable? A table with this data must be included in supplemental material.
Change CDX+ for CDX+.
Line 38: in vivo.
Section 2.2: Please add more details besides the cited reference.
Were dead cells excluded from the analysis in flow cytometry? Why did the authors not include the absolute numbers of cells? The authors must include the gating strategy in the supplemental figures.
Figure 3A: CD8+.
The authors must include the rationale for only analyzing IFN-gamma and MIP- 1 beta in the text. Also, it is crucial to measure other pro-inflammatory cytokines, at least by ELISA.
Lines 321 and 334: in silico.
Comments on the Quality of English Language
Minor grammar editing is required.
Author Response
Reviewer 2
Clinical data from patients must be included. Are differences in sex, age, or another
variable? A table with this data must be included in supplemental material.
Answer: Thank you for your suggestion.
We have added the patient characteristics as the supplemental Table 1 and mentioned it
on page 2, line 90.
Change CDX+ for CDX+.
Answer: Thank you for your comment.
We have changed all of CDX+ to CDX+.
Line 38: in vivo.
Answer: Thank you. We changed it to italic.
Section 2.2: Please add more details besides the cited reference.
Answer: Thank you for your comment.
Briefly, DNA was extracted from PBMCs and 50 ng of DNA was subjected to PCR with
sequence-specific primer sets for HLA-A*02 (#292 and #170) and for an internal control
(#63 and #64), using AmpliTaq Gold 360 master mix (Applied BioSystems, Tokyo,
Japan). The cycling program included denaturation at 96°C for 25 sec, annealing at 70°C
for 45 sec, and extension at 72°C for 45 sec for the first 5 cycles; denaturation at 96°C
for 25 sec, annealing at 65°C for 50 sec, and extension at 72°C for 45 sec for the next 21
cycles; and denaturation at 96°C for 25 sec, annealing at 55°C for 60 sec, and extension
at 72°C for 120 sec for the final 4 cycles. The PCR products were electrophoresed in a
2% agarose gel. The product sizes are 557 bp and 796 bp, respectively.
We added the above on page 3, line 100.
Were dead cells excluded from the analysis in flow cytometry?
Answer: Thank you for the comment.
We excluded dead cells by cytogram gating in the flow cytometry analysis, but did not
use a detection reagent for dead cells. Instead, in the functional T cell avidity assay, we
minimized background noise by subtracting the percentage of positive cells in the noantigen
sample. This is noted on page 4, line 174.
Why did the authors not include the absolute numbers of cells?
Answer: We agree with the reviewer’s comment.
We used 1x106 cells for tetramer staining, 2x106 cells for each CTL function assay, and
1x106 cells for the T cell avidity assay. We have added these cell numbers to the text.
The authors must include the gating strategy in the supplemental figures.
Answer: Thank you for your comment.
We have added the gating strategy in supplemental figure 1.
Figure 3A: CD8+.
Answer: Thank you for your suggestion. We have added this indication, “in CD8+ gate”,
above the plots, as Figure 3A presents flow cytometry data gated on CD8+ cells.
The authors must include the rationale for only analyzing IFN-gamma and MIP- 1 beta
in the text. Also, it is crucial to measure other pro-inflammatory cytokines, at least by
ELISA.
Answer: Thank you for your comment.
We chose IFN-γ and MIP-1β as the cytokine and chemokine, respectively, because our
preliminary experiments indicated that these proteins were the most sensitive. This is
included in the methods section on page 3, line 137, as “these molecules were the most
sensitive in preliminary examinations.” As the reviewer pointed out, it is important to
measure other cytokines. Unfortunately, we did not have enough cells from the same
patients to perform these assays.
Lines 321 and 334: in silico.
Answer: Thank you for your suggestion. We changed it to italic in line 334 and 347.

Reviewer 3 Report
Comments and Suggestions for Authors
The manuscript entitled "HLA-A*24 increases the risk of HTLV-1-associated myelopathy despite reducing HTLV-1 proviral load", by Tanaka et al brings up the results of an interesting experiment that detected an inverse association of specif Tax CTL and proviral load, but with an increased risk with HAM/TSP. In general, the manuscript is well written and scientifically sound. I have just a question to authors: why among 132 patients included in the study only 8 were included in the CTL assays?
In addition, why to use almost 50% of references from last century? Most of original data are currently outdated. For instance, the reference 3 (Uchiyama), from 1977 is cited to support the statement that only 1-2% of patients develop HTLV-associated diseases, which is not true, according to the current knowledge. I strongly suggest a revision and updating of references list.
Comments on the Quality of English Language
Minor English editing to adequate some grammar problems.
Author Response
Reviewer 3
I have just a question to authors: why among 132 patients included in the study only 8
were included in the CTL assays?
Answer: Thank you for your question.
We chose samples from 8 patients with over 0.5% CTL to obtain accurate results in the
functional assays. This is included on page 3, line 138, as “For precise evaluation of
CTL function, we initially selected HAM/TSP patients with a high frequency of Tax
11–19- or 301–309-specific CTLs exceeding 0.5%.” Additionally, since a relatively
high number of cells were needed for these assays, we selected only the 8 available
patients.
In addition, why to use almost 50% of references from last century? Most of original
data are currently outdated. For instance, the reference 3 (Uchiyama), from 1977 is cited
to support the statement that only 1-2% of patients develop HTLV-associated diseases,
which is not true, according to the current knowledge. I strongly suggest a revision and
updating of references list.
Answer: Thank you for your comment. In response to the reviewer's comment, we have
replaced 9 references with recently reported papers and updated the prevalence of
HTLV-1-related diseases to: 'The lifetime risk of developing adult T-cell leukemia
(ATL) in people with HTLV-1 is estimated to be 2–7%, whereas the risk of developing
HTLV-1-associated myelopathy/tropical spastic paraparesis (HAM/TSP) is 0.25–3%”
on page 1, line 41.

Round 2
Reviewer 2 Report
Comments and Suggestions for Authors
The authors have addressed almost all the suggestions.
Comments on the Quality of English LanguageMinor editing is required.
Reviewer 3 Report
Comments and Suggestions for Authors
The question/comment were properly addressed by authors